# Cardiac dopamine D1 receptor triggers ventricular arrhythmia in chronic heart failure

Toshihiro Yamaguchi[1,14], Tomokazu S. Sumida [1,2,14], Seitaro Nomura[1,3], Masahiro Satoh [3,4], Tomoaki Higo[5], Masamichi Ito[1], Toshiyuki Ko[1], Kanna Fujita[1], Mary E. Sweet[6], Atsushi Sanbe[7], Kenji Yoshimi[8], Ichiro Manabe[9], Toshikuni Sasaoka[10], Matthew R. G. Taylor[6,11], Haruhiro Toko[1,12], Eiki Takimoto[1], Atsuhiko T. Naito [13✉] & Issei Komuro[1✉]

Pathophysiological roles of cardiac dopamine system remain unknown. Here, we show the role of dopamine D1 receptor (D1R)-expressing cardiomyocytes (CMs) in triggering heart failure-associated ventricular arrhythmia. Comprehensive single-cell resolution analysis identifies the presence of D1R-expressing CMs in both heart failure model mice and in heart failure patients with sustained ventricular tachycardia. Overexpression of D1R in CMs disturbs normal calcium handling while CM-specific deletion of D1R ameliorates heart failure-associated ventricular arrhythmia. Thus, cardiac D1R has the potential to become a therapeutic target for preventing heart failure-associated ventricular arrhythmia.

[1] Department of Cardiovascular Medicine, Graduate School of Medicine, The University of Tokyo, Tokyo, Japan. [2] Department of Neurology and Immunobiology, Yale School of Medicine, New Haven, CT, USA. [3] Genome Science Division, Research Center for Advanced Science and Technologies, The University of Tokyo, Tokyo, Japan. [4] Department of Cardiovascular Medicine, Chiba University Graduate School of Medicine, Chiba, Japan. [5] Department of Cardiovascular Medicine, Osaka University Graduate School of Medicine, Osaka, Japan. [6] Human Medical Genetics and Genomics, University of Colorado, Boulder, CO, USA. [7] Department of Pharmacotherapeutics, School of Pharmacy, Iwate Medical University, Morioka, Japan. [8] Department of Neurophysiology, Juntendo University Graduate School of Medicine, Tokyo, Japan. [9] Department of Disease Biology and Molecular Medicine, Graduate School of Medicine, Chiba University, Chiba, Japan. [10] Department of Comparative and Experimental Medicine, Brain Research Institute, Niigata University, Niigata, Japan. [11] Cardiovascular Institute and Adult Medical Genetics Program, University of Colorado, Boulder, CO, USA. [12] Department of Advanced Translational Research and Medicine in Management of Pulmonary Hypertension, The University of Tokyo, Tokyo, Japan. [13] Department of Physiology, Faculty of Medicine, Toho University, Tokyo, Japan. [14] These authors contributed equally: Toshihiro Yamaguchi, Tomokazu Sumida. ✉email: atsuhiko.naito@med.toho-u.ac.jp; komuro-tky@umin.ac.jp

Heart failure has become a compelling clinical issue due to its increase in morbidity and mortality, and affects >24 million patients worldwide[1]. Approximately half of the patients with heart failure die of cardiac sudden death caused by ventricular arrhythmia. Therefore, prevention of life-threatening ventricular arrhythmias, as well as cardiac dysfunction is critically important for improving the prognosis of the patients with heart failure[2–4].

Dopamine is an endogenous catecholamine that functions as a neurotransmitter in the central nervous system, but it also works as an autocrine and paracrine factor in the nonneuronal systems. Due to its inotropic effect on myocardium and natriuretic effect, dopamine has been widely used as one of the therapeutic drugs for the patients with acute heart failure[5,6]. A decade ago, however, a large clinical study demonstrated that the prognosis of patients with chronic heart failure was worsened by the treatment with dopamine receptor agonists and suggested that ventricular arrhythmia might be related to the poor prognosis[7,8]. This study indicated a pathological role of D1R signaling in heart failure; however, molecular mechanisms by which the activation of D1R might increase the risk of patients with heart failure have not been explored yet. Here, we show that D1R-expressing CMs play critical roles in triggering heart failure-associated ventricular arrhythmia and clarified its molecular mechanisms.

## Results

**Heterogenous appearance of cardiac D1R in failing hearts**. To identify molecules associated with the progression of heart failure, we searched for genes whose expression levels were progressively increased throughout the development of heart failure. To this end, we made a mouse model of pressure overload-induced heart failure by transverse aortic constriction (TAC) operation (Supplementary Table 1) and performed RNA sequencing (RNA-seq) analysis using heart tissues at the early and the late phases of heart failure (Fig. 1a). Among the genes whose expression levels were progressively upregulated or downregulated in the failing heart, we focused on *Drd1*, which encodes a D1R, because only *Drd1* was significantly upregulated in failing hearts among the catecholamine receptors expressed in the heart as has been observed in another publicly available RNA-seq data (Supplementary Fig. 1a). Quantitative PCR (qPCR) confirmed that expressions of *Drd1*, but not other dopamine receptor genes, were increased (Fig. 1b and Supplementary Fig. 1b). Moreover, the expression of *Drd1* was strongly related with that of *Nppa* and *Nppb*, which are well-established markers of cardiac hypertrophy and heart failure (Supplementary Fig. 1c). We next examined the spatial expressions of *Drd1* in the failing heart by using single-molecule fluorescence in situ hybridization (smFISH) with our previously developed algorithm, which allows us to assess mRNA levels of targeted genes in CMs[9] (Fig. 1c). Expressions of *Drd1* were identified in a limited number of CMs of sham-operated mice, and the number of *Drd1*-expressing CMs was increased during the progression of heart failure induced by pressure overload (Fig. 1d). Heterogeneous expression of *Drd1* mRNA and the increase in the number of *Drd1*-expressing CMs were also confirmed by single-cell RNA-seq of the failing heart[10] (Fig. 1e). Furthermore, genetic labeling of D1R-expressing cells using transgenic mice expressing LacZ in D1R-expressing cells[11] further confirmed the presence of small numbers of D1R-expressing cells after TAC operation (Fig. 1f and Supplementary Fig. 2a–c). These results collectively indicate that expressions of D1R were increased partly in a limited number of CMs, but not uniformly in many CMs, resulting in heterogenous and patchy expressions of D1R in the failing hearts.

**Deletion of cardiac D1R ameliorates ventricular arrhythmia**. To examine the roles of cardiac D1R in the pathophysiology of

heart failure, we generated mice with CM-specific deletion of *Drd1* gene by crossing *αMHC-Cre* mice with *Drd1*flox/flox mice (*αMHC-Cre/Drd1*fl/fl; Fig. 2a). Deletion of *Drd1* in the CMs of *αMHC-Cre/Drd1*fl/fl mice was confirmed at both DNA and mRNA levels (Fig. 2b, c). There was no significant difference in the left ventricular (LV) function of *αMHC-Cre/Drd1*fl/fl mice and *αMHC-Cre* mice, even after the TAC operation (Supplementary Fig. 3a and Supplementary Table 2). A previous clinical trial reported that the treatment with dopamine analog increased the incidence of cardiac sudden death related with arrhythmia in the patients with severe heart failure[7]. We therefore hypothesized that the increase in the number of D1R-expressing CMs might contribute to the arrhythmogenesis in the failing heart. To test our hypothesis, we first examined the frequency of ventricular arrhythmia by a surface electrocardiogram (ECG) under anesthesia. Intravenous infusion of a mixture of dopamine and caffeine was used to increase the frequency of ventricular arrhythmia (Supplementary Fig. 3b). The frequency of premature ventricular contraction (PVC) was significantly lower in TAC-operated *αMHC-Cre/Drd1*fl/fl mice compared with TAC-operated *αMHC-Cre* mice (Fig. 2d). We also analyzed the frequency of PVC in awake mice by telemetric ECGs. In this case, we used intraperitoneal injection of a mixture of dopamine and caffeine to increase the frequency of ventricular arrhythmia. In accordance with the results of the surface ECG under anesthesia, the frequency of PVC in *αMHC-Cre/Drd1*fl/fl mice was significantly lower than in *αMHC-Cre* mice (Supplementary Fig. 3c, d). Moreover, *αMHC-Cre/Drd1*fl/fl mice showed lesser mortality compared with *αMHC-Cre* mice (Fig. 2e) despite the comparable reduction of contractile function (Supplementary Fig. 3a).

**Overexpression of cardiac D1R triggers ventricular arrythmia**. We next investigated whether the overexpression of *Drd1* gene is sufficient to trigger ventricular arrhythmia in vivo. We crossed *αMHC-tTA* knock-in mice and *TRE-D1R/lacZ* transgenic mice to create *αMHC-tTA/TRE-D1R* mice. In this transgenic mouse, D1R was overexpressed specifically in CMs, but the overexpression is silenced by treatment with doxycycline (Fig. 3a and Supplementary Fig. 4a). The frequency of drug (dopamine and caffeine)-induced PVC was significantly higher in *αMHC-tTA/TRE-D1R* mice compared with control mice (Fig. 3b). Furthermore, PVC frequency in *αMHC-tTA/TRE-D1R* mice was decreased by addition of doxycycline that reduces D1R expression, while doxycycline administration to wild mice didn't affect the frequency of PVCs in wild-type mice (Supplementary Fig. 4b, c). In addition, cardiac function was not altered with overexpression of D1R in CMs (Supplementary Fig. 4d and Supplementary Table 3). These results collectively suggest that the upregulation of cardiac D1R is both necessary and sufficient for triggering life-threatening ventricular arrhythmia during heart failure independent of cardiac function.

To better understand the molecular mechanisms by which D1R overexpression triggers cardiac arrhythmia, we focused on calcium ($Ca^{2+}$) handling; abnormal $Ca^{2+}$ handling in CMs is one of the major causes of arrhythmogenesis in heart failure[12,13]. Overexpression of D1R elicited arrhythmia and increased the frequency of $Ca^{2+}$ sparks in CMs (Fig. 3c–e and Supplementary Fig. 4e, f). Myocardial $Ca^{2+}$ handling is tightly controlled via multiple mechanisms, including the ryanodine receptor 2 (RyR2) that is central to the regulation of induction of spontaneous $Ca^{2+}$ sparks. RyR2 hyperphosphorylation is known to contribute on arrhythmogenesis in the context of heart failure. Protein kinase A (PKA) and $Ca^{2+}$/calmodulin-dependent protein kinase II (CaMKII) are well known to activate RyR2 in CMs[14,15]. Considering neuronal D1R can

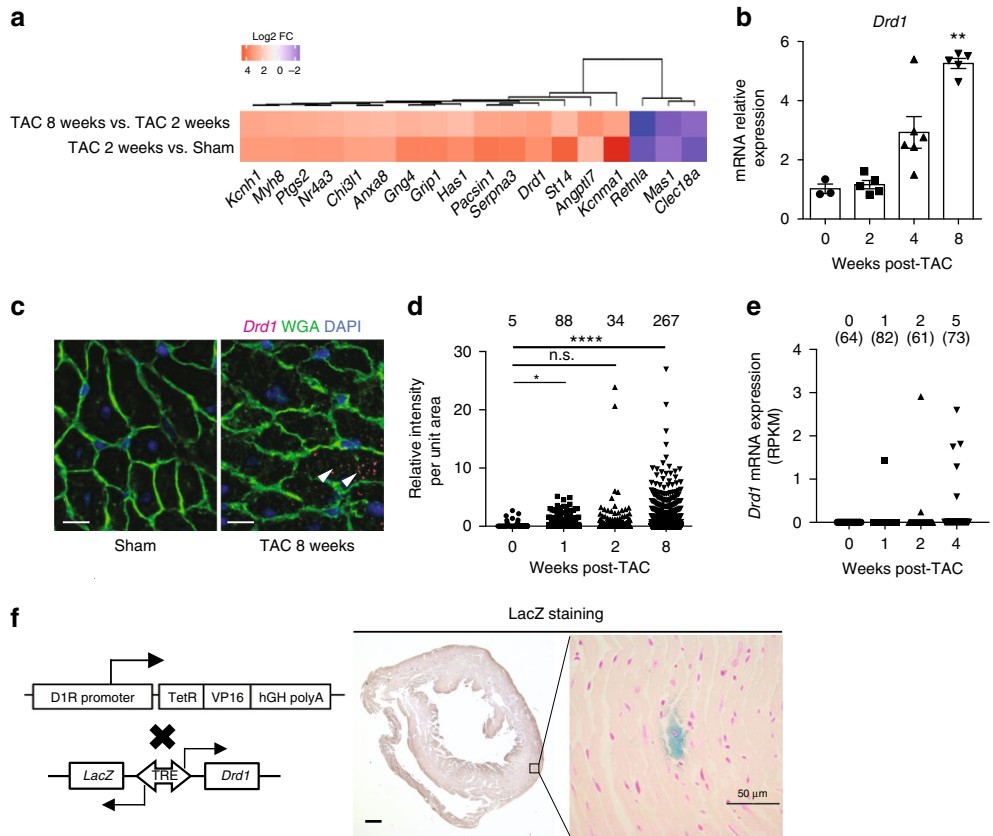

**Fig. 1 Appearance of D1R-expressing cardiomyocytes in failing heart. a** Heatmap showing the genes whose expression was increased or decreased during the progression of heart failure. Tissue samples were collected from sham-operated, 2 and 8 weeks after TAC-operated hearts (sham, $n = 2$; 2 weeks, $n = 3$; and 8 weeks, $n = 1$). **b** Bar graph showing relative expression of *Drd1* in heart tissues after TAC operation analyzed by qPCR (0 week: sham-operated, $n = 3$; 2 weeks, $n = 5$; 4 weeks, $n = 6$; and 8 weeks, $n = 5$). Data are shown as mean and s.e.m. Statistical significance was determined by Kruskal–Wallis test followed by Dunn's multiple comparisons test; $**p = 0.0077$ versus 0 week. **c** Representative images of single-molecule fluorescence in situ hybridization against *Drd1* (magenta). Heart tissue was collected 8 weeks after the sham operation (sham) or TAC operation (TAC 8 weeks). Tissue samples were stained with WGA (green), and DAPI (blue). Arrowheads indicate the positive staining of *Drd1* mRNA. Scale bars, 10 μm. **d** Dot plots showing relative fluorescent intensity of *Drd1* in CMs normalized by the area of each cells. A total of 680, 1935, 2344, and 2180 CMs from one mouse for each time point were analyzed. Statistical significance was determined by Kruskal–Wallis test followed by Dunn's multiple comparisons test; $*p = 0.021$; $****p < 0.0001$ versus sham (0 week); n.s. not significant. The numbers above the dot plots indicate the number of CMs whose normalized intensity was more than one. **e** Dot plots showing the expression of *Drd1* in CMs isolated from the heart tissue of sham- and TAC-operated mice at 1, 2, and 4 weeks after the operation ($n = 2$ each) evaluated by single-cell RNA-seq. The numbers above the dot plots indicate the number of *Drd1*-expressing CMs followed by the number of CMs analyzed. **f** Left: generation of D1R LacZ reporter mice (*D1R-tTA/TRE-D1R/lacZ* mice). tTA tetracycline transactivator, TRE tetracycline responsive element, TetR tetracycline repressor, VP16 virion protein 16, hGH human growth hormone. Right: representative image of LacZ staining of heart tissue from TAC-operated mice (4 weeks after surgery). Scale bar 500 μm (left) and 50 μm (right). Nucleus is counterstained in magenta and LacZ-stained cells in blue.

activate both PKA[16,17] and CaMKII[18,19], we hypothesized that aberrant induction of cardiac D1R might increase RyR2 phosphorylation that consequently impairs $Ca^{2+}$ handling and triggers ventricular arrhythmia. Immunoblot analyses revealed that phosphorylations at two major sites of RyR2, serine 2808 (S2808) and serine 2814 (S2814), were increased in the heart of *αMHC-tTA/TRE-D1R* mice compared with control mice (Fig. 3f). These data indicate a potential mechanism by which cardiac D1R drives dysregulated $Ca^{2+}$ handling via hyper activation of RyR2 in failing hearts.

**Increased expression of cardiac D1R in patients with heart failure**. We finally assessed the expression of D1R in patients with heart failure. RNA-seq analyses of whole heart tissues from patients with cardiomyopathy of various etiology[20,21] showed that D1R expression was upregulated in human failing hearts (Supplementary Fig. 5a). Furthermore, our ad hoc analysis on previous cohort[21] demonstrated that among patients with

cardiomyopathy, the expression levels of cardiac D1R tended to be higher in patients with a history of ventricular tachycardia and/or with an implantable cardioverter defibrillator device (Supplementary Fig. 5b). We next analyzed the transcriptome of the CMs from the patients by single-cell RNA-seq. There was no CM that expressed D1R among the 72 CMs examined in the patient with normal cardiac function. In contrast, D1R-expressing CMs were identified in 9 out of 11 patients with heart failure (Fig. 3g). Moreover, the number of D1R-expressing CMs tended to be higher in the heart failure patients with a clinical history of ventricular tachycardia (Fig. 3g). These observations support our findings in animal models and suggest a role for D1R-expressing CMs in the pathogenesis of ventricular arrhythmias in patients with heart failure. These findings in human patients may also explain the result of the previous clinical study showing that dopamine analogs worsened the prognosis of the patients, with chronic heart failure by increasing the incidence of cardiac sudden death related with arrhythmia[7].

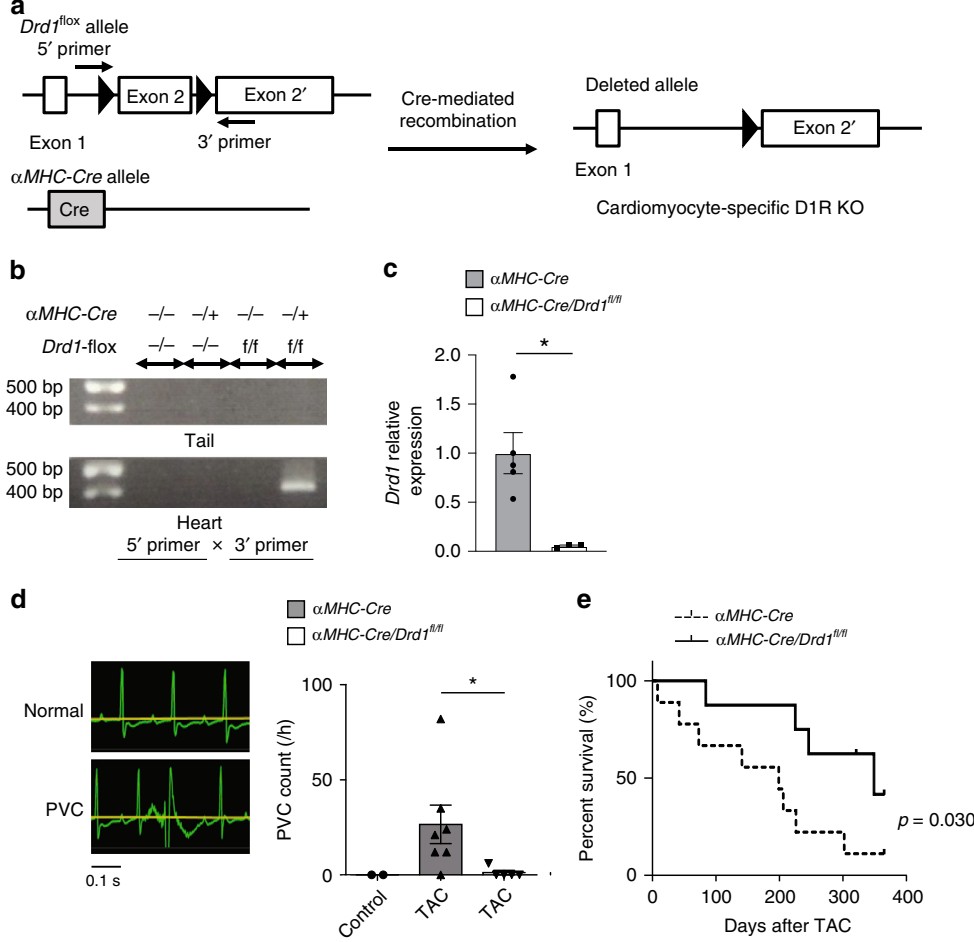

**Fig. 2 Deletion of cardiac D1R ameliorates heart failure-associated ventricular arrhythmia. a** Generation of cardiac-specific D1R-deficient mice and the design of the PCR primer. KO knockout. **b** Representative results of the genotyping PCR from both tail and heart tissues. Genomic DNA was extracted from the mice with indicated genotypes. **c** Bar graph showing the expression of *Drd1* in TAC-operated hearts of *αMHC-Cre* or *αMHC-Cre/Drd1*^fl/fl mice assessed by qPCR; (TAC-operated *αMHC-Cre*, n = 5; and TAC-operated *αMHC-Cre/Drd1*^fl/fl, n = 3). Data are shown as mean and s.e.m. Statistical significance was determined by Mann–Whitney *U* test; *p = 0.036 versus control. **d** Left: representative surface electrocardiogram recordings of the normal sinus rhythm (normal) and the premature ventricular contraction (PVC). Right: bar graph showing the frequency of drug-induced PVC in *αMHC-Cre* and *αMHC-Cre/Drd1*^fl/fl mice under anesthesia. The frequency of PVC in *αMHC-Cre* mice before TAC operation is shown as a control; (control, n = 2; *αMHC-Cre* (TAC), n = 7; and *αMHC-Cre/Drd1*^fl/fl (TAC), n = 5). Data are shown as mean and s.e.m. Statistical significance was determined by Kruskal–Wallis test followed by Dunn's multiple comparisons test; *p = 0.023; *αMHC-Cre* (TAC) versus *αMHC-Cre/Drd1*^fl/fl (TAC). **e** Survival curves comparing post-TAC survival of *αMHC-Cre* and *αMHC-Cre/Drd1*^fl/fl mice (*αMHC-Cre*, n = 8; and *αMHC-Cre/Drd1*^fl/fl, n = 9). Statistical analysis was performed by Gehan–Breslow–Wilcoxon test; *p = 0.030.

## Discussion

In this study, we showed that the expression of D1R was upregulated and correlated with ventricular arrhythmia in the failing heart of both murine models and human patients. Although a few papers have suggested the existence of D1R in human and rat hearts[22–24], there has been no report addressing the physiological and pathophysiological function of cardiac D1R. Our results provide the first direct experimental evidence to our knowledge for a role of cardiac D1R in heart failure. Results from single-cell RNA-seq, smFISH imaging, and genetic labeling all indicated that upregulation of D1R in failing hearts was not a global event that take place in all CMs but, rather, was observed only in a limited number of CMs. These results are consistent with the fact that increased D1R in failing hearts did not affect global cardiac function, but instead, the induction of PVCs. We have recently reported the spatio-temporal heterogeneity of gene expression among CMs during the development of heart failure[10]. Our present study provides an example by which cellular heterogeneity may play a critical role in the pathogenesis of cardiac

diseases, highlighting the importance of single-cell approaches for dissecting the molecular mechanism of diseases. The results of murine experiments strongly suggest that increased cardiac D1R is responsible for driving ventricular arrhythmia specifically in the context of heart failure. We observed a trend for heart failure patients with higher D1R expression to have more clinical events for ventricular arrhythmia (Supplementary Fig. 5b). A similar trend was also observed in our single-cell RNA-seq cohort (Fig. 3g). Although our human results did not reach the statistical significance possibly because of the small sample size, we believe it deserves clinical attention. Further studies with more heart failure patients are needed to verify the association between cardiac D1R induction and ventricular arrhythmogenesis in human.

Altogether, these findings demonstrate the role of D1R-expressing CMs in triggering heart failure-associated ventricular arrhythmia. The dopamine–dopamine receptor system has been extensively studied in the field of neuroscience and its aberrant regulation plays essential roles in various neurological

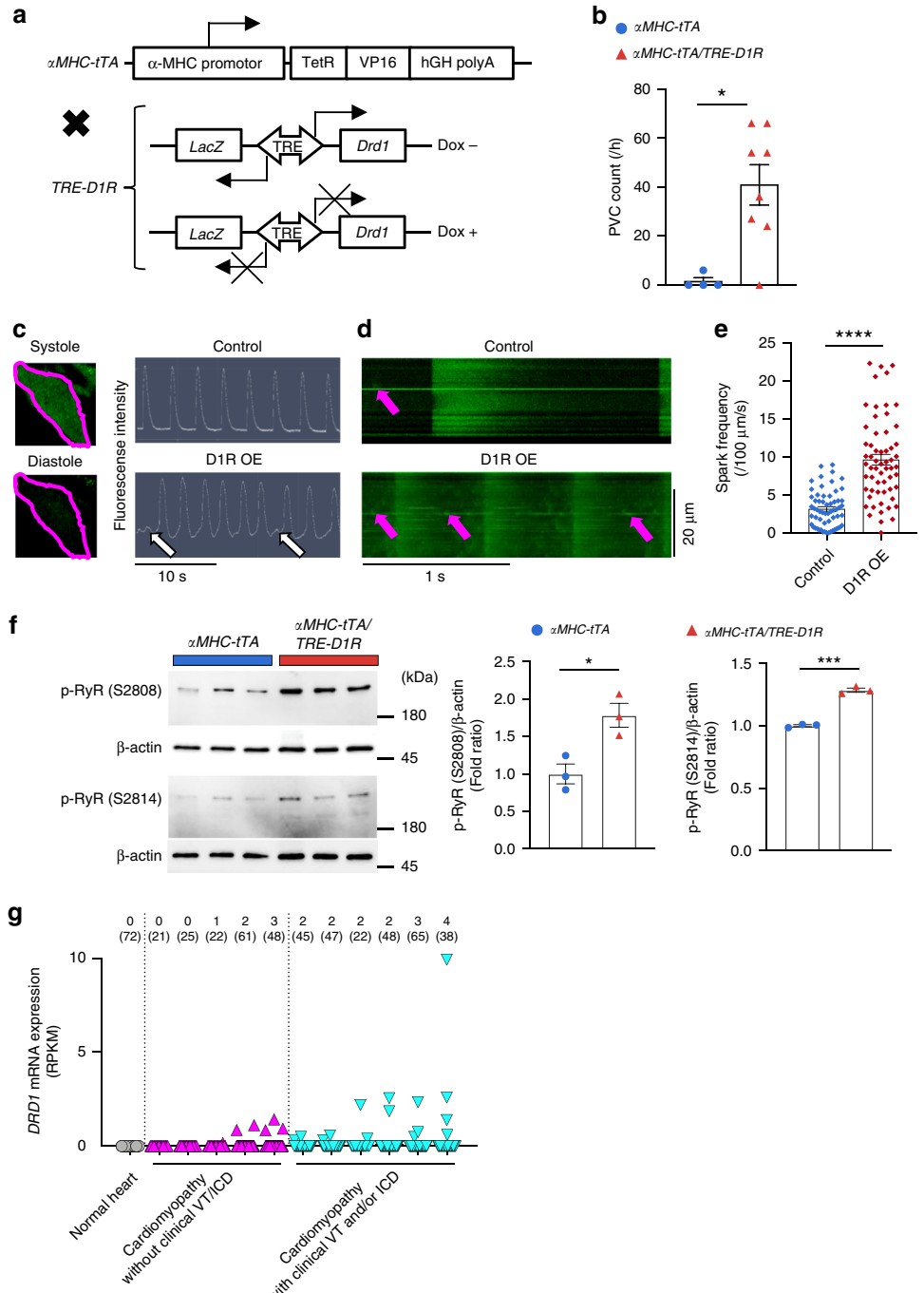

disorders[25–27]. Future approaches to target only peripheral D1R, ideally cardiac D1R, would be required for preventing ventricular arrhythmias in patients with severe and chronic heart failure.

## Methods

**Animal models.** All the animal experiments were approved by the University of Tokyo Ethics Committee for Animal Experiments and strictly adhered to the guidelines for animal experiments of the University of Tokyo (Approved Number P13-141 and P16-102). All of wild-type C57BL/6 mice were purchased from CLEA Japan. C57BL/6-Drd1a^tm1a(KOMP)Wtsi mice were purchased from the KOMP repository and crossed with C57BL/6-Tg(CAG-Flpe)2 Arte to delete the neomycin resistance gene and obtain Drd1^flox/+ heterozygous mice. Drd1^flox/flox homozygous mice were crossed with αMHC-Cre mice that were purchased from Jackson Laboratory (stock#009074, stock name: STOCK Tg (Myh6-cre)1Jmk/J) for the generation of CM-specific Drd1 knockout mice. Deletion of Drd1 allele in the heart was examined by detection of floxed allele using primers 5′-AAG GCGCATAACGATACCAC-3′ and 5′-TTTCTTCTTGAGCTGGCTTTT-3′. B6.

Cg-Drd1a^tm1(tTA)Mok mice (D1R-tTA mice) were crossed with B6.Cg-Tg(TRE-Drd1/lacZ)1Mok mice (TRE-D1R/lacZ mice) to obtain the transgenic mice that express LacZ in D1R-expressing cells. αMHC-tTA mice were crossed with TRE-D1R/LacZ transgenic mice for generating the CM-specific D1R-overexpressing mice whose expression of D1R could be naturally induced together with the expression of LacZ reference, and this induction is reversed by administration of doxycycline through the drinking water. αMHC-tTA/TRE-D1R mice were treated with 0.2 g/L doxycycline (Sigma D-9891) in tap water containing 1% sucrose for 2 weeks. Doxycycline-containing water was shielded from light and was refreshed twice per week. D1R-tTA mice[11], TRE-D1R/lacZ mice[11], and αMHC-tTA mice[28] were established as described previously.

**TAC operation and echocardiography.** Transverse aortic constriction (TAC) operation was performed to 8–12-week-old male mice. Transgenic mice were all backcrossed in C57BL/6 strain for more than ten generations and age-matched pairs of mice were used in the study. We also used littermates to minimize the age differences. The transverse aorta was ligated with a 7-0 silk suture paralleled with a 26 or 27-gauge blunt needle, which was removed after the ligation. Sham-operated

**Fig. 3 D1R overexpression in cardiomyocytes triggers aberrant calcium handling. a** Generation and *Drd1* gene expression of CM-specific D1R-overexpressing mice with or without doxycycline (Dox) treatment. **b** Bar graph showing the frequency of drug-induced PVC in *αMHC-tTA* mice (control) and *αMHC-tTA/TRE-D1R* mice (Tg; control, n = 4; and Tg, n = 8). Data are shown as mean and s.e.m. Statistical significance was determined by Mann–Whitney *U* test; *p = 0.020 versus control. **c** Left: representative images of a CM stained with calcium-sensitive dye (green: Cal520®) during systole and diastole. Margin of the cell is traced in magenta. Right: representative calcium transients in LacZ-overexpressing (Control) and D1R-overexpressing CMs (D1R OE). Arrows indicate abnormal calcium transient. **d** Representative line scan images of calcium transients of LacZ-overexpressing (control) and D1R-overexpressing (D1R OE) CMs. Arrows indicate calcium sparks. **e** Bar graph showing the frequency of calcium sparks in LacZ-overexpressing (control) and D1R-overexpressing (D1R OE) CMs (control, n = 60; and D1R OE, n = 60). CMs were collected from 20–30 neonatal rats and pooled samples were used for the experiments. Data are shown as mean and s.e.m. Statistical significance was determined by Mann–Whitney *U* test; ****p < 0.0001 versus control. **f** Left: representative immunoblots of phosphorylated-RyR between *αMHC-tTA* and *αMHC-tTA/TRE-D1R* mice. p-RyR (S2808), phosphorylated RyR2 at Ser 2808; p-RyR (S2814), phosphorylated RyR2 at Ser 2814. Right: bar graph showing quantitative evaluation of immunoblotting for phosphorylated-RyR (S2808 and S2814) fold ratio between *αMHC-tTA* and *αMHC-tTA/TRE-D1R* mice (n = 3, each). Data are shown as mean and s.e.m. Statistical significance was determined by unpaired *t*-test; S2808 *p = 0.020 and S2814 ***p = 0.0001 versus control. **g** Dot plots showing the expression of *DRD1* gene in human CMs evaluated by single-cell RNA-seq. Data from each patient are plotted individually. The patients with cardiomyopathy were divided into two groups according to the histories of clinical VT. The numbers above the dot plots indicate the number of *DRD1*-expressing CMs followed by the number of CMs analyzed; (patient with normal cardiac function (normal heart; n = 1), patients with cardiomyopathy without clinical VT and ICD (n = 5), and with clinical VT and/or ICD (n = 6)).

animals were subjected to similar surgical procedure without aortic constriction and used as control. TAC operation was performed to the mice weighing 22–27 g at the age of 8–12-week old. Surgeon was not informed about the genotype of the mice. The mice that died within 1 week after the operation were excluded from the analysis. Transthoracic echocardiography was performed to conscious mice with a Vevo 2100 imaging system (Visualsonics Inc.). M-mode echocardiographic images were obtained from a longitudinal view to analyze the LV size and LV contraction.

**Surface ECG**. Sham- and TAC-operated mice were anesthetized with 1.75% iso-flurane. Electrodes were placed to the limbs and ECG was continuously monitored with a Vevo 2100 system (VisualSonics Inc.). A mixture of dopamine (2 mg/mL) and caffeine (12 mg/mL) was injected using a micro syringe pump (ESP-64, Eicom, Kyoto, Japan) to induce ventricular arrhythmia. The flow rate of injection was increased every 10 min and the number of ventricular arrhythmias during each flow rate was counted.

**Telemetric ECG**. A mouse-specific wireless telemetry probe TA11ETA-F10 (Data Sciences International (DSI), St Paul, MN, USA) was implanted to TAC-operated mice under anesthesia and the data were obtained continuously from conscious mice. To induce arrhythmia, a mixture of dopamine (20 mg/kg) and caffeine (120 mg/kg) was injected intraperitoneally. The number of ventricular arrhythmias was counted semiautomatically using Data Insights (PONEMAH v5.20, DSI). Ventricular arrhythmias were first defined from representative waveform using provided protocols (protocol dv hr missed beats and protocol dv hr change). Then the number of ventricular arrhythmias during the drug-free phase (spontaneous phase) and after drug injection for 1 h (drug-induced phase) was analyzed.

**Isolation of neonatal rat CMs**. Neonatal rat CMs were prepared from the ventricles of newborn Wistar rats[29]. The ventricular part of the heart was quickly excised and washed in ice-cold PBS. Digestion of the ventricles was performed seven times with 0.05% trypsin-EDTA at 37 °C for 10 min. Supernatant from the first digestion was discarded, and those from the other digestion steps containing CMs were collected. Non-CMs were separated from CMs by plating the cells to a culture dish in Dulbecco's modified Eagle's medium (DMEM, Gibco) containing 10% fetal bovine serum (FBS) and the cells were cultured at 37 °C for 50 min. Nonattached cells were collected and used as CMs. CMs were cultured in DMEM containing 10% FBS and 25 μM cytosine β-D-arabinofuranoside hydrochloride (Ara-C, Sigma) was added to inhibit the proliferation of non-CMs.

**LacZ staining**. Perfusion fixation was performed to anesthetized mice by using fixation buffer containing 0.2% glutaraldehyde, 1% formaldehyde, and 0.02% NP-40 in PBS. Tissue samples (heart and brain) were then dissected and postfixed in the fixation buffer for 2 h at room temperature. After washing in PBS, samples were incubated in a staining solution containing potassium ferrocyanide (5 mM), potassium ferricyanide (5 mM), MgCl₂ (2 mM), 0.01% NP-40, 0.01% sodium deoxycholate, and 1 mg/mL X-gal (Wako) in PBS for 8 h at 37 °C. Heart samples were then embedded in paraffin and sectioned at 5 μm thickness. Nuclei were visualized by Nuclear Fast Red (Sigma) and the images were taken using Leica DM 2500 LED microscope.

**Single-molecule fluorescence in situ hybridization**. smFISH was performed with an RNAscope® (Advanced Cell Diagnostics, Newark, CA) according to the manufacturer's instruction[9]. The target probe against mouse *Drd1a* mRNA (NM_010076.3, bp 444-1358, ACD# 406491-C2) was designed using a previously

described protocol[30]. Frozen sections in 10 μm thickness were fixed in PBS containing 4% paraformaldehyde for 15 min at 4 °C, dehydrated sequentially by 50, 70, and 100% ethanol for 5 min each at room temperature, and treated with protease for 30 min at room temparature. The tissue samples were incubated with target probes for 2 h at 40 °C, followed by a series of signal amplification and co-staining with fluorescein-conjugated wheat germ agglutinin (WGA; 1:100) to detect cell boundaries. The sections were counterstained with DAPI. Images were obtained by IN Cell Analyzer 6000 (GE Healthcare) and the quantification of RNA molecules was performed by In Cell Developer Toolbox 1.9.1 (GE Healthcare). The total intensity of labeled *Drd1* mRNA per single CMs that was surrounded by WGA was measured.

**Adenoviral overexpression of D1R**. Recombinant adenovirus vectors were generated for overexpression of D1R in CMs using AdEasy Adenoviral Vector System (Agilent Technologies INc., Santa Clara, CA) according to the manufacturer's instruction. Complementary DNA clone for *Drd1* was purchased from TransOMIC Technologies and fused with FLAG-tag. Genetic Ca²⁺ indicator GCaMP8 (a generous gift from Professor Junichi Nakai, Brain Science Institute, Saitama University) was coexpressed with D1R by using inserting internal ribosomal entry site. LacZ was used as a control for D1R overexpression (Supplementary Fig. 4e). The titer of the adenovirus was determined by AdEasy viral titer kit (Agilent Technologies) according to the manufacturer's instructions[31]. Adenovirus was infected to CMs at 3 days after the isolation at 25 m.o.i. for 6 h. Cells were stained with commercially available anti-Tag antibody[32–34] (1:1000 dilution; PM020; Medical & Biological Laboratories Co., Ltd., Nagoya, Japan) to confirm the overexpression of tagged D1R (Supplementary Fig. 4f).

**Ca²⁺ imaging**. Ca²⁺ imaging was performed as previously described with modifications[35]. An LSM 510 scanning system (Zeiss, Jena, Germany) equipped with ×63 oil immersion objective was used for the imaging of Ca²⁺. Temperature was held at 37 °C and CO₂ concentration at 5% by CO₂ Module S and Temp Module S system (Zeiss). Fibronectin-coated glass bottom dish (Matsunami) was used for live cell imaging. Ca²⁺ transient was recorded 2 days after the infection. Cells were incubated with 5 μM of Cal520®AM (AAT) diluted in DMEM containing 10% FBS for 1 h, in order to gain further fluorescence strength. A line scan mode was used for the recording of Ca²⁺ sparks. Fluorescence dye was excited by 488 nm line of an argon laser and emission signals over 505 nm were collected. The confocal pinhole was set to 96 μm and detector gain was set at 1000. Line scan images were acquired at sampling rate of 0.66 ms per line, along the longitudinal axis of the cell. Each line comprised 1024 pixels spaced at 0.1 μm intervals. After a sequential scanning, we generated a two-dimensional image of 1024 × 10,000 lines and counted the number of Ca²⁺ sparks. The spatially averaged Ca²⁺ transients were obtained by plane scan image of cell area.

**Immunoblot analysis**. Immunoblot analysis was performed as previously described[36]. Heart tissue was homogenized in ice-cold RIPA buffer containing protease inhibitor (cOmplete Protease Inhibitor Cocktail, Merck)) and phosphatase inhibitor (PhosSTOP, Sigma). Supernatant was collected after centrifugation at 15,000 × *g* and an aliquot containing equal amount of protein was mixed with loading buffer (125 mM Tris-HCl pH 6.8, 30% glycerol, 10% SDS, and 0.6 M DTT). Without heat denaturation, protein sample was subjected to SDS–PAGE and transferred to PVDF membrane (Thermo Fisher Scientific). Specific antibodies against phospho-RyR2(Ser2808) (1:1000 dilution), phospho-RyR2(Ser2814) (1:1000 dilution; Badrilla)[36–39], β-actin (1:2500 dilution; Invitrogen)[40–42] were used as primary antibodies. Horseradish peroxidase-conjugated anti-rabbit IgG antibody

(1:3000 dilution; Cell Signaling Technology) and anti-mouse IgG antibody (1:3000 dilution; Cell Signaling Technology) were used as secondary antibodies. Immunoreactive signals were detected with ECL prime Western Blotting Detection System (GE Healthcare).

**qPCR analysis**. Total RNA was extracted using TRIzol reagents (Invitrogen) for heart tissues and cultured cells according to manufacturer's instructions. RNA samples were subjected to DNase treatment to remove genomic DNA and reverse-transcribed using QuantiTect® Reverse Transcription Kit (QIAGEN). qPCR was performed using Universal Probe Library (UPL, Roche) and THUNDERBIRD® Probe qPCR Mix (TOYOBO). Relative expression levels of the target genes were normalized to the expression of internal control gene using comparative Ct method. Primer sequences and the corresponding UPL numbers were designed with online program provided by Roche. Full list of primer sequences are available in Supplementary Table 5.

**RNA-seq analysis of whole heart**. Total RNA was isolated as described above. The quality of the RNA was assessed by nanodrop measurement. RNA-seq libraries were prepared by Truseq sample purification kit (Illumina). The libraries were sequenced on Genome Analyzer II (Illumina), and the reads were uniquely aligned to the mouse genome (mm9) by CASAVA software version 1.7 (Illumina). The reads per kilobase of exon per million mapped reads (RPKM) of each gene were calculated based on the length of the gene and the read counts mapped to the gene.

**Isolation of CMs and single-cell RNA-seq analysis of heart failure model mice**. Single-CM transcriptomes were obtained using mice model of pressure overload-induced heart failure[10]. CMs were isolated from the heart of adult mice using the Langendorff perfusion apparatus[43–45]. Briefly, the heart was quickly excised and retrogradely perfused through the aorta with $Ca^{2+}$-free buffer (130 mM NaCl, 5.4 mM KCl, 0.5 mM $MgCl_2$, 0.33 mM $NaH_2PO_4$, 22 mM D-glucose, 25 mM HEPES, 30 mM 2,3-butanedione monoxime, pH 7.4) for 2 min at 4 mL/min, then with collagenase solution (1 mg/mL collagenase type 2 (Worthington), 0.05 mg/mL protease type 14 (Sigma Aldrich), and 100 μM $CaCl_2$ in 0-$Ca^{2+}$ buffer) for 10 min at 3 mL/min. The whole system was maintained at 39 °C. Following perfusion, hearts were placed in 35-mm dish containing 100 μM $Ca^{2+}$ buffer (100 μM $CaCl_2$ in 0-$Ca^{2+}$ buffer) with 2 mg/mL bovine serum albumin and gently minced with microscissors. The cell suspension was filtered through 200 μm mesh to remove undigested tissue and CMs were collected by centrifugation at $100 \times g$ for 2 min. The supernatant was discarded and CMs were resuspended in medium (NaCl 130 mM, KCl 5.4 mM, $MgCl_2$ 0.5 mM, $NaH_2PO_4$ 0.33 mM, D-glucose 22 mM, HEPES 25 mM, FBS 0.2%, pH 7.4) containing a low concentration of $Ca^{2+}$ (0.1 mM). Rod-shaped living CMs were collected immediately after isolation and incubated in lysis buffer. Smart-seq2 protocol[10] generated the single-cell RNA-seq libraries. A total of 280 single-CM transcriptomes of sham-operated, 1, 2, and 4 weeks after TAC-operated mice were analyzed. RPKM values were calculated with reads mapped to the nuclear genome using DEGseq (version 1.8.0).

**Human sample collection and RNA-seq analysis of whole heart tissue**. All experiments undergone at the University of Colorado Hospital were approved by the Colorado Multiple Institutional Review Board (COMIRB, protocol 01-568), and written consent had been obtained from all participants. The protocols of sample collection and RNA-seq analysis of whole heart tissue were previously described[21]. The history of ICD implantation or sustained ventricular tachycardia events in 12 months prior to heart transplantation was collected. These clinical histories were obtained by independent operator who was blinded to the result of RNA-seq analysis about *DRD1*.

**Human sample collection and single-cell RNA-seq analysis**. All experiments were approved by the ethics committee of the University of Tokyo (Approved Number G-10032). All procedures were conducted according to the Declaration of Helsinki, and written informed consent was obtained from all participants. Heart tissue was obtained immediately after death due to noncardiac cause (one patient with normal cardiac function) or during LV assist device surgery or heart transplantation (11 patients with severe heart failure). The history of sustained ventricular tachycardia was collected from clinical records of our hospital, patient referral document from other hospitals, or cardiac monitoring records of implantation device. These clinical histories were obtained by independent operator who was blinded to the result of single-cell RNA-seq analysis. Demographics and characteristics of each patient were shown in Supplementary Table 4. Immediately after the collection of the heart tissue, rod-shaped living CMs were isolated, and then incubated in lysis buffer. The method of single-cell RNA-seq analysis using the Smart-seq2 protocol was previously described[10]. Single-CM RNA-seq data has been deposited in the Gene Expression Omnibus with the accession code GSE 95143.

**Statistical analysis**. Statistical analyses were performed in Prism 6.07 (GraphPad). All statistics are described in figure legends. In general, for two experimental comparisons, two-tailed unpaired or paired (Supplementary Fig. 4b) Student's

*t*-test was used. For multiple comparisons, one-way ANOVA and Kruskal–Wallis test was used as indicated. Data were represented as the average of indicated number of biologically independent samples and s.e.m. unless indicated. The statistical significance is represented by asterisks corresponding to *$p < 0.05$, **$p < 0.01$, ***$p < 0.001$, and ****$p < 0.0001$.

**Reporting summary**. Further information on research design is available in the Nature Research Reporting Summary linked to this article.

## Data availability

The authors declare that all data are available in the article file and Supplementary information files, or available from the authors upon reasonable request. The Source data underlying Figs. 1a, b, d, e, 2c, d, e and 3b, e, f, g, Supplementary Figs. 1a–c, 3a, b, d, 4a–d and 5a, b, and Supplementary Tables 1–3 are provided as a Source data file. RNA-seq data from whole heart of mice have been deposited in GSE 29446 and GSE 95143. Single-cell RNA-seq data from CMs of mice and human have been deposited in GSE 95143. RNA-seq data from human heart have been deposited in GSE 116250 and GSE 46224. Source data are provided with this paper.

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

## Acknowledgements
We gratefully acknowledge the technical assistance of N. Yamanaka and M. Hayashi. We also thank Dr. S. Okuda and Dr. M. Yano (Yamaguchi University Graduate School of Medicine) for giving technical advice for immunoblotting of phospho-RyR2, and K. Shiina, T. Fujita, and Dr. H. Aburatani (Research Center for Advanced Science and Technologies, The University of Tokyo) for next-generation sequencing support. A sincere thank you to Dr. B. Nelson for his conscientious proofreading. This work was supported by grants from the Japan Foundation for Applied Enzymology and a Grant-in-Aid for Early-Career Scientists (to T.Y.). AMED under Grant Number JP19am0101122, a Grant-in-Aid for Scientific Research (B), The Cell Science Research Foundation, SEN-SHIN Medical Research Foundation, and Miyata Cardiac Research Promotion Foundation (to A.T.N.). A Grant-in-Aid for Scientific Research (B) (to S.N.), a Grant-in-Aid for Scientific Research (A) (to I.K.), and AMED-CREST under Grant Number JP20gm0810013, JP20ek0109406, JP20km0405209, JP20bm0704026, JP20gm6210010, JP20ek0109487, JP20ek0109440, JP20ek0210141, and JP19bm0804010 (to S.N. and I.K.).

## Author contributions
T.Y., T.S., E.T., A.T.N., and I.K. conceived the project, designed the study, and interpreted the results. T.Y. and T.S. performed in vivo and in vitro experiments with the help of M.I., and A.T.N. S.N., and M.S. collected single cells and generated the single-cell sequencing data. M.S. and T.Y. performed single-cell RNA-seq, single-molecule RNA FISH, and immunohistochemistry experiments. T.Y., T.S., and H.T. performed immunoblot analysis. A.S. generated αMHC-tTA mice. T.S. generated the transgenic mice that express LacZ in D1R-expressing cells and TRE-D1R mice. T.K. and K.F. collected the human samples and clinical histories of the patients at the University of Tokyo. M.E.S. and M.R.G.T provided the clinical information with the previous study21. T.H., A.T.N., M.I., A.S., K.Y., I.M., T.S., and I.K. provided experimental and analytical support. T.Y., T.S., E.T., and A.T.N. wrote the manuscript with feedback from all authors.

## Competing interests
The authors declare no competing interests.
