## [Peer Review File · Nature Communications]

Reviewers' comments:

Reviewer #1 (Remarks to the Author):

General comments: This report deals with the role of the cardiac dopamine D1 receptor (D1R) in the pathogenesis of cardiac arrhythmia in heart failure. The combination of studies in rodents and humans is a strength of this report. There is a decrease in premature ventricular contractions (PVCs) in mice with cardiomyocyte-specific deletion of *Drd1* subjected to thoracic aorta constriction (TAC) (Figure 1g). However, the PVCs induced by dopamine and caffeine do not seem to be decreased by cardiomyocyte-specific deletion of *Drd1* (Figure 2a). The blood pressures of the mice were not measured (Card Electrophysiol Clin. 2015 Jun;7(2):207-20).

Specific concerns:

A. Major:

1. The statement in (line 14) "Increased expression of *Drd1*, but no other dopamine receptor genes was [sic] confirmed by quantitative PCR (qPCR)" needs proof. The D1R, D3R, and D4R have been reported to be expressed in the heart (Pharmacol Ther. 2019 Jul 9:107392), as well as the D2R and D5R in the human (Heart Vessels. 2010 Sep;25(5):432-7 (reference 17)) and rat hearts (Circulation. 2012 Dec 11;126(24):2859-69).

2. Figure 2b is cited to indicate that the prevention of the cardiomyocyte-specific deletion of D1R decreases the number of PVCs. However, it seems that the number of PVCs with doxycycline treatment is greater than that in β MHC-tTA.

3. Supplementary Fig. 5b. The failure to find a significant difference in cardiac DRD1 mRNA in patients with history of sustained ventricular tachycardia and/or with an implantable cardioverter defibrillator device than patients without ventricular tachycardia weakens the linkage between increased cardiac DRD1 expression and ventricular tachycardia.

3. Supplementary Table. The average expression of DRD1 mRNA in patients with no tachycardia or nonsustained tachycardia can be higher (44 years old, 40 years old#2 and #3, and 38 years old) than several patients with sustained ventricular tachycardia).

B. Minor:

1. References:

a. Reference 5 deals with ibopamine which is an agonist at both α -adrenergic and dopamine receptor (Clin Cardiol. 1995 Mar;18(3 Suppl I):117-21).

b. Reference 6 deals with the treatment of shock, not heart failure.

2. Grammar. The manuscript needs editing for grammar and syntax.

3. Formatting:

- a. LacZ is upside down in Figure 1f. Sometimes, there is a space between Lac and Z.

Reviewer #2 (Remarks to the Author):

This manuscript investigates the role of dopamine receptor D1 (D1R)-expressing cardiomyocytes in triggering heart failure-associated ventricular arrhythmias. The authors show that D1R-expressing cardiomyocytes are found in mice subject to TAC induced heart failure as well as heart failure patients with sustained ventricular arrhythmias. Overexpression and deletion of D1R is generated in mouse models and correlates directly with the incidence of ventricular arrhythmias. The authors conclude that abnormal calcium handling in cardiomyocytes results from D1R expression during heart failure and predisposes to ventricular arrhythmia. This study reveals a potentially interesting role for the dopamine receptor in heart failure associated ventricular arrhythmias. The data does present significant data that expression levels of D1R rise during heart failure and correlates with an increased frequency of mild ventricular arrhythmias. Concerns are listed below.

Specific Points:

1. Of the 18 genes that qualified as significantly changed in the RNA sequencing why was the DR1 chosen and why were others excluded. Is currently very arbitrary.
2. Data is presented regarding mouse models of heart failure. Criteria is listed for excluding animals but none regarding those included for data analysis. Were hearts analyzed for structural changes consistent with TAC, were other parameters such as ejection fractions, fractional shortening and chamber dimensions monitored to determine the success and progression of heart failure in experimental animals.
3. In accordance with the above issue (ques.2) does the overexpression of DR1 accelerate TAC in α MHC2 tTA/TRE-D1R mice and slow in the DR1 knockdown model? Could the increased incidence of arrhythmias observed be related to the extent of HF rather than altered calcium dynamics?
4. It appears that DR1 expression is localized to a small percentage of myocytes with even fewer cells showing significantly elevated levels at 8 wks post-TAC. Although DR1 levels correlate with increased PVCs there is not any evidence that demonstrates that these few DR1 positive cells are the initiating source of these PVCs. Although one can draw the conclusion that DR1 overexpression results in more calcium dysfunction (sparks) there is no direct evidence in the WT TAC model that this is the case.

5. No data is presented regarding the mechanism of increased sparks in the DR1 overexpression model. Is the level of overexpression similar, more, less to what is seen in the average ventricular myocytes after 8 wks TAC. Is the arrhythmia burden significantly higher in the overexpression model assuming that all and not only a fraction of cells are now overexpressing DR1 or less because of the now more homogeneous (since heterogeneous distribution has been proposed as mechanism for arrhythmia) nature of the substrate.

Reviewer #3 (Remarks to the Author):

In this short communication, Yamaguchi, Sumida and colleagues reveal a potential role for the dopamine receptor D1R in triggering ventricular arrhythmia during heart failure. Evidence is gathered and condensed from an impressive array of techniques including single cell RNA sequencing of CMs in both a mouse TAC model and human heart failure patients, smFISH, loss and gain of function mouse genetic models and in vivo surface and wireless telemetric ECG. The work and findings are novel and of high clinical relevance. However, there are several areas of concern, outlined below.

Major:

1) The in vivo mouse experiments are incompletely described, to an extent that sometimes precludes interpretation of data.

- What are the ages of mice used for all experiments? This is only specified for TAC/sham operated mice, described as between 8-12 weeks old. Are males and females used?

- How were these mice allocated into experimental groups to avoid confounding factors of age (8-12 weeks is a considerable range), gender (if applicable), and batch effects due to different litters?

- The method for doxycycline treatment is not described. In fact, the methods section suggests "tetracycline" administration, via drinking water (p10 lines 15-16). Please clarify. What was the drug, supplier, dose and duration of treatment in this experiment?

- Figure 1f, how many weeks post-surgery is this?

2) Linked to the above, there are statistical concerns regarding the methodology used to compare between groups in single cell/slice comparisons, particularly because it is unclear how many actual mice were used per condition, and how many cells/slices came from each mouse.

- Figure 1d, how many mice per timepoint? How many cells analysed per mouse? Why are different numbers of cells analysed for each timepoint, and only 680 for control? When performing the statistical test, should the mean values for each mouse not be calculated first, and then the statistical test performed on these mean values, in order to give a biologically informed comparison?

- Figure 1e, how many mice per condition? How many cells from each mouse?

- Figure 1f, as above for 1d. How many mice per condition? How many slices analysed per mouse? Were mean values per mouse used for the stats test. Also, why is a bar chart used here instead of a dot-plot?

- Figure 2e, was this experiment reproduced with a second batch of neonatal CMs? Or are all data obtained from one biological experiment? n=9 and n=10 seem small if this refers to numbers of cells analysed. Were the cells randomly selected, was the experimenter blinded to the experimental condition?

3) The study relies upon mRNA expression to infer expression of the D1R protein receptor in CMs. This requires validation of D1R protein expression by immunohistochemistry. There appear to be several antibodies available for this. Authors might co-stain with a CM-specific marker to demonstrate CM-specific protein expression of D1R in mouse and human control and heart failure tissue.

4) It is claimed that figures 1d and 1f demonstrate D1R expression in CMs. However, how do the authors discount the possibility that some +ve stained cells are non-CMs? For example, D1R expression in vasculature is known. Authors should either quantify only D1R+ staining in cells that are co-stained with a CM-specific marker, or, clearly acknowledge the caveat that some staining here may arise from non-CM cells.

5) Doxycycline is a known cardioprotective agent [PMIDs 24104875, 31274354]. There is therefore concern that reduced ventricular arrhythmia following DOX administration (Fig 2b) could be confounded by the effect of DOX itself, rather than reduced D1R expression. An ideal solution could be to repeat this experiment using a Tet-ON system (aMHC-rtTA). However, alternatively, authors might demonstrate that DOX administration alone does not rescue ventricular arrhythmia in wild-type mice after TAC (i.e. figure 1g experimental setup).

Minor:

1) Were there any differences in D1R KO body weights, or heart weights?

Also, were there differences in heart contractile function, wall thickness, heart weight or body weight in the D1R overexpression model? Note that dosage of the caffeine-dopamine intravenous drug infusion was not adjusted for body weight.

2) Figure 1f: On average, how many total CMs are there per slice? 0.3 LacZ+ cells per slice (TAC condition) seems low. Does this correspond to the proportion of D1R+ CMs identified by scRNAseq (from Figure 1e, for 4 week post-TAC, this proportion is $5/73 = 6.8\%$)?

The representative image in figure 1f and supplementary figure 2c suggest that fewer than 6.8% of CMs are LacZ+. Can the authors comment on this discrepancy?

3) Supplementary Figure 3c: No error bars, was this $n=1$? Also, was this measured using an un-operated mouse? If so, it's surprising that such a clear difference exists, given that in un-operated mice, almost no CMs are expected to express D1R. Can the authors comment on this?

4) P4 Line 14 "Increased expression of *Drd1*, but no other dopamine receptor genes, was confirmed by quantitative PCR (qPCR)." Only qPCR for D1R and D2R are shown – what about others?

5) Please show primer sequences used for qPCR.

Response to reviewers:

We thank the editor and the three reviewers for their interest in our paper and thoughtful comments. We have addressed all comments in a point-by-point fashion as below. Based on the comments and suggestions, we have performed additional experiments and analyses and cleared up confusion over our text as well as expanded on topics requested by Reviewers.

Specifically, we agree with the editor and reviewer that the mechanism of how D1R overexpression affects calcium dynamics is an important question that remains to be addressed and have now performed several additional experiments to answer this question. All changes within the manuscript are highlighted with yellow markings. Overall, we think that the revised manuscript contains more data than the original version and the plot outline becomes more clear and solid. Our responses to reviewer comments are marked with blue text. We wish to thank all the reviewers for their attention to details and for their kind and informative comments.

Reviewer #1 (Remarks to the Author):

General comments: This report deals with the role of the cardiac dopamine D1 receptor (D1R) in the pathogenesis of cardiac arrhythmia in heart failure. The combination of studies in rodents and humans is a strength of this report. There is a decrease in premature ventricular contractions (PVCs) in mice with cardiomyocyte-specific deletion of Drd1 subjected to thoracic aorta constriction (TAC) (Figure 1g). However, the PVCs induced by dopamine and caffeine do not seem to be decreased by cardiomyocyte-specific deletion of Drd1 (Figure 2a). The blood pressures of the mice were not measured (Card Electrophysiol Clin. 2015 Jun;7(2):207-20).

Specific concerns:

A. Major:

1. The statement in (line 14) "Increased expression of Drd1, but no other dopamine receptor genes was [sic] confirmed by quantitative PCR (qPCR)" needs proof. The D1R, D3R, and D4R have been reported to be expressed in the heart (Pharmacol Ther. 2019 Jul 9:107392), as well as the D2R and D5R in the human (Heart Vessels. 2010 Sep;25(5):432-7 (reference 17)) and rat hearts (Circulation. 2012 Dec 11;126(24):2859-69).

Response: We thank the reviewer for the suggestion. We had already performed qPCR for determining the expression of *Drd3*, *Drd4*, and *Drd5* in mice samples, and surprisingly we could not detect their gene expression by qPCR, (also confirmed by our RNA-seq data and others) (GSE29466 and ERP000771) (Figure R1-1). With regard to *Drd2*, qPCR data was shown in the original manuscript supplementary figure 1a and the result was consistent with our RNA-seq data. We also analyzed the human heart RNA-seq data (GSE46224 and GSE116250) to check if *DRD3*, *DRD4*, and *DRD5* were detectable. Unexpectedly, *DRD4* was expressed at the highest level among the dopamine receptor family in the human heart. *DRD3* genes were

barely detected and *DRD5* was expressed at lower level compared to the others (Figure R1-1). These data suggest that the expression levels of dopamine receptors are slightly different between human and mice specifically for *DRD4*, however, we would like to emphasize that this difference doesn't have an impact on our conclusion, because *DRD4* belongs to the dopamine D2-like family (*Drd2*, *Drd3*, and *Drd4*), which is coupled to the Gi/Go family of G protein that has an opposite function to dopamine D1-like family (*Drd1* and *Drd5*). The qPCR data have been included in revised manuscript Supplementary Figure 1a and the qPCR primer sequence for *Drd3-5* was listed in Supplementary table3.

Figure R1-1: Expression of dopamine receptors in mice and human heart tissue. Bulk RNA-seq data of heart tissue with TAC or heart failure condition (three for mice data and two for human data) were analyzed and the expression value of dopamine receptors (*Drd1-5* (mouse), *DRD1-5* (human)) were shown.

2. Figure 2b is cited to indicate that the prevention of the cardiomyocyte-specific deletion of *D1R* decreases the number of PVCs. However, it seems that the number of PVCs with doxycycline treatment is greater than that in α MHC-*tTA*.

Response: We thank the reviewer for the comment. With regard to remaining higher PVCs even after DOX treatment, we found that *Drd1* gene expression was significantly lower than before DOX treatment but still higher than control mice (Figure R1-2 and Supplementary Figure 4a). This relatively higher expression of *Drd1* in DOX treated α MHC-*tTA/TRE* mice could explain the higher frequency of PVCs in these mice. In addition, α MHC-*tTA/TRE* mice have been exposed to the forced expression of cardiac D1R before DOX treatment, and we think this pre-exposure to D1R overexpression should be considered as a confounding factor and could explain the remaining low-grade PVCs even after cancelling D1R overexpression by DOX treatment. To avoid confusion, we decided to split Figure 2b into two figures. One with the un-paired data representing difference between α MHC-*tTA* mice and α MHC-*tTA/TRE* mice (Figure 2b), and the other representing the paired comparison before and after DOX treatment in

αMHC-tTA/TRE mice (Supplementary Figure 4b). By integrating together, we argue that the overexpression of cardiac D1R is necessary and sufficient for induction of PVCs. We revised the text and figure as indicated above.

Figure R1-2: Cardiac *Drd1* expression in α MHC-tTA/TRE mice before and after DOX treatment. qPCR analysis of *Drd1* expression in heart tissue from control and α MHC-tTA/TRE mice (D1R overexpression mice) before and after DOX administration. *Drd1* expression was highly induced in α MHC-tTA/TRE mice before DOX administration. Two weeks of DOX treatment suppressed *Drd1* expression significantly, but not completely.

3. *Supplementary Fig. 5b. The failure to find a significant difference in cardiac DRD1 mRNA in patients with history of sustained ventricular tachycardia and/or with an implantable cardioverter defibrillator device than patients without ventricular tachycardia weakens the linkage between increased cardiac DRD1 expression and ventricular tachycardia.*

Response: We thank the reviewer for the suggestion. We agree that the trend showing higher D1R expression in heart failure patients prone to ventricular arrhythmia did not reach statistically significant difference; however, we do not think the result “weaken” the linkage between increased cardiac *Drd1* expression and ventricular arrhythmia. Rather, we think the trend, which is consistent with the findings in animal experiment, supports our hypothesis and encourages us and the physicians/researchers in the field for further investigation. Unlike the data from experimental animals, interpretation of clinical data is sometimes difficult especially when the number of patients is limited and the background etiologies of the patients are diverse. Future clinical study with more patient number or with more well-designed selection criteria would be required to verify the link between dopamine-dopamine receptor system and ventricular arrhythmia in heart failure patients. We added sentences about the interpretation of clinical data in page 10, line 5-17. We thank the reviewer for raising the important point.

3. *Supplementary Table. The average expression of DRD1 mRNA in patients with no tachycardia or nonsustained tachycardia can be higher (44 years old, 40 years old#2 and #3, and 38 years old) than several patients with sustained ventricular tachycardia).*

Response: We thank the reviewer for the comment. It is well known that ventricular arrhythmias develop as a result of different electro-physiological mechanisms, including abnormal impulse formation (triggered activity, automaticity) and conduction disturbances (reentry), which could involve aberrant regulation of intracellular potassium or sodium ions as well as that of calcium

ions. Deranged mitochondrial metabolism and the structural change of heart tissue such as fibrosis could also play significant roles. Given that the data was obtained from the heart failure patients with different etiologies, it is not surprising that some patients with sustained VT revealed low levels of *DRDI* mRNA expression. In addition, as almost all patients were on anti-arrhythmia and anti-heart failure medications, it is very possible that some patients with high levels of *DRDI* mRNA expression did not develop sustained VT thanks to efficacious treatments. Although we could not observe statistically significant but slight tendency in the relationship between *DRDI* and lethal arrhythmias likely due to the limited number of patient samples, we believe this is an important piece of information and would like to clarify this in the future study. Accordingly, we have provided additional information on medications (including one additional scRNA-seq analysis of DCM patient) and incorporated some of the aforementioned discussion in the text.

B. Minor:

1. References:

a. Reference 5 deals with ibopamine which is an agonist at both α -adrenergic and dopamine receptor (Clin Cardiol. 1995 Mar; 18(3 Suppl 1):117-21).

Response: It is true that ibopamine also works as an agonist for α -adrenergic receptor. However, at therapeutic dose, it is reported that ibopamine works only on dopamine receptor². Thus, the outcome of this study (Reference 5) should be interpreted as an effect of dopamine receptor agonist, not as an α -adrenergic receptor agonist.

b. Reference 6 deals with the treatment of shock, not heart failure.

Response: We removed the reference accordingly.

2. Grammar. The manuscript needs editing for grammar and syntax.

Response: We corrected grammar and syntax and the final manuscript is checked by native English speaker.

3. Formatting:

a. LacZ is upside down in Figure 1f. Sometimes, there is a space between Lac and Z.

Response: We apologize for the poor editing quality. We corrected the figures as pointed out by the reviewer.

Reviewer #2 (Remarks to the Author):

This manuscript investigates the role of dopamine receptor D1 (DIR)-expressing

cardiomyocytes in triggering heart failure-associated ventricular arrhythmias. The authors show that DIR-expressing cardiomyocytes are found in mice subject to TAC induced heart failure as well as heart failure patients with sustained ventricular arrhythmias. Overexpression and deletion of DIR is generated in mouse models and correlates directly with the incidence of ventricular arrhythmias. The authors conclude that abnormal calcium handling in cardiomyocytes results from DIR expression during heart failure and predisposes to ventricular arrhythmia. This study reveals a potentially interesting role for the dopamine receptor in heart failure associated ventricular arrhythmias. The data does present significant data that expression levels of DIR rise during heart failure and correlates with an increased frequency of mild ventricular arrhythmias. Concerns are listed below.

Specific Points:

1. Of the 18 genes that qualified as significantly changed in the RNA sequencing why was the DR1 chosen and why were others excluded. Is currently very arbitrary.

Response: We thank the reviewer for the comment. We focused on *Drd1* because most of us are physicians and use dopamine and dopamine receptor blockers as a drug in our clinical practice, but never thought the level of dopamine receptor changes in heart failure. In fact, we were also interested in the change in *Ptgs2* level because we also use COX2 inhibitors in our clinical practice. However, as the beneficial effect of COX2 inhibitors in preventing cardiovascular diseases is already established, we started to focus on *Drd1* to seek for the novelty. We were also interested in the contrasting expression of dopamine receptor genes and β -adrenergic receptor genes: the expression of *Drd1* and *Adrb2* become upregulated whereas *Drd2* and *Adrb1* become downregulated (Supplementary Figure 1a). Notably, the clinical trials with ibopamine on chronic heart failure patients not only failed but the risk of mortality was increased in the cohort of ibopamine users¹, suggesting the detrimental effect of activating dopamine receptors in heart failure. Whereas, a potential pathogenic role of dopamine receptors in heart failure have been overshadowed by the adrenergic receptor response, and the molecular mechanisms by which the activation of dopamine receptors increases the risk of heart failure events have not been studied well. These clinically relevant questions and the largely unexplored area of research on dopamine receptors in the heart prompted us to investigate the molecular details of dopamine receptors in the context of mouse and human heart failure. We added a comment why we focused on the change in *Drd1* expression (page 4, line 12 - page 5, line 1) and Supplementary Figure 1a in the revised version.

2. Data is presented regarding mouse models of heart failure. Criteria is listed for excluding animals but none regarding those included for data analysis. Were hearts analyzed for

structural changes consistent with TAC, were other parameters such as ejection fractions, fractional shortening and chamber dimensions monitored to determine the success and progression of heart failure in experimental animals.

Response: We thank the reviewer for the comment. Yes, we observed structural changes consistent with TAC. Increase in ventricular chamber dimension and decrease in left ventricular fractional shortening are observed in mice after TAC operation. The results of transthoracic echocardiography as well as body weight, heart weight and lung weight, are now included in the revised manuscript (Supplementary Table 1-1).

3. In accordance with the above issue (ques.2) does the overexpression of DR1 accelerate TAC in α MHC2 tTA/TRE-D1R mice and slow in the DR1 knockdown model? Could the increased incidence of arrhythmias observed be related to the extent of HF rather than altered calcium dynamics?

Response: We thank the reviewer for the comment. CM specific knockout of *Drd1* gene did not affect the severity of heart failure (Supplementary Figure 3d) but significantly decreased the incidence of arrhythmia after TAC. We therefore do not think that incidence of arrhythmia is related to the extent of heart failure but altered calcium dynamics due to D1R knockout. We did not perform TAC operation to D1R-overexpressing mice because the purpose of using *Drd1*-overexpressing mice was to observe the “pure” effect of D1R-overexpression, not affected by the severity of heart failure, on the incidence of arrhythmia. In fact, *α MHC-tTA/TRE-D1R* mice exhibited normal cardiac function (Supplementary Figure 4d). Above results and the basal characteristics of CM specific D1R knockout mice and overexpressing mice are now included in the revised manuscript (page 6, line 4-6 (Supplementary Table 1-2), and page 7, line 11-12 (Supplementary Table 1-3)).

4. It appears that DR1 expression is localized to a small percentage of myocytes with even fewer cells showing significantly elevated levels at 8 wks post-TAC. Although DR1 levels correlate with increased PVCs there is not any evidence that demonstrates that these few DR1 positive cells are the initiating source of these PVCs. Although one can draw the conclusion that DR1 overexpression results in more calcium dysfunction (sparks) there is no direct evidence in the WT TAC model that this is the case.

Response: We thank the reviewer for the comment. We agree with the reviewer that there is no direct evidence showing that few D1R expressing cells are the initiating source of PVC in TAC-treated mice. However, our loss-of-function model (cardiac specific D1R knockout mice) clearly indicated that limiting the appearance of those few D1R-expressing cells suppressed the incidence of PVC, indirectly suggesting that few D1R expressing cells are the initiating source

of PVC in TAC model mice. We believe that our approach using genetically engineered mouse model (cardiac specific D1R knockout mouse as in Figure 1g) is one of the most straightforward ways to address the question raised by the reviewer.

5. No data is presented regarding the mechanism of increased sparks in the D1R overexpression model. IS the level of overexpression similar; more, less to what is seen in the average ventricular myocytes after 8 wks TAC. Is the arrhythmia burden significantly higher in the overexpression model assuming that all and not only a fraction of cells are now overexpressing D1R or less because of the now more homogeneous (since heterogeneous distribution has been proposed as mechanism for arrhythmia) nature of the substrate.

Response: We thank the reviewer for the comment. Although we did not experimentally assess the copy number of mRNA in the heart of WT mice with TAC operation and of CM specific D1R-overexpressing mice, we assume that the expression level would be much higher in the latter, because of the strength of α MHC promoter. The frequency of PVC was almost comparable in WT mice with TAC and D1R-overexpressing mice (30 ± 5 and 40 ± 8 , respectively). However, given that the incidence of PVC in mice with heart failure were supposed to be affected by other factors that do not involve D1R, it is assumed that arrhythmia burden is significantly higher in D1R overexpressing mice.

We understand our cardiac specific overexpression model is not perfect. We showed that D1R overexpression induce aberrant calcium handling in cardiomyocytes and increase the incidence of PVC in the heart but the expression level and the pattern of expression do not completely reproduce the situation in the failing heart. However, given the technical difficulty in controlling both distribution and amount of D1R expression in cell-type specific manner, we think our approach would be the best available at the current moment.

As pointed out by the reviewer, we also investigated for the downstream mechanisms by which D1R overexpression induce aberrant Ca^{++} leakage. We showed that the phosphorylation levels of RyR2 (ryanodine receptor 2) S2808 and S2814 are both increased in the heart of mice overexpressing D1R (Figure 2f). As phosphorylation of RyR2 at these sites contribute to arrhythmogenesis through altering Ca^{++} handling in the failing heart, it is suggested that aberrant phosphorylation of RyR2 mediate the expression of D1R and arrhythmogenesis. The data has been included in revised Figure 2f and we added the description in the text (page 8, line 1-12).

Reviewer #3 (Remarks to the Author):

In this short communication, Yamaguchi, Sumida and colleagues reveal a potential role for the dopamine receptor D1R in triggering ventricular arrhythmia during heart failure. Evidence is

gathered and condensed from an impressive array of techniques including single cell RNA sequencing of CMs in both a mouse TAC model and human heart failure patients, smFISH, loss and gain of function mouse genetic models and in vivo surface and wireless telemetric ECG. The work and findings are novel and of high clinical relevance. However, there are several areas of concern, outlined below.

Major:

1) The in vivo mouse experiments are incompletely described, to an extent that sometimes precludes interpretation of data.

- What are the ages of mice used for all experiments? This is only specified for TAC/sham operated mice, described as between 8-12 weeks old. Are males and females used?

Response: As well as TAC experiment, we used 8-12 weeks old male mice for all of the experiments. The information has been included in Supplementary Table 1-1 and Life science reporting summary.

- How were these mice allocated into experimental groups to avoid confounding factors of age (8-12 weeks is a considerable range), gender (if applicable), and batch effects due to different litters?

Response: We controlled the gender bias by just using male mice. The bias driven by age was also controlled by performing experiment with same littermates. In each set of experiment, the gap of age was less than two weeks, but there were small variations in the age across the experiments. To avoid the confusion, we revised the description in method section. Since the results are represented as replicated data acquired from different set of littermates, it can be interpreted as non-biased data for age and gender.

- The method for doxycycline treatment is not described. In fact, the methods section suggests “tetracycline” administration, via drinking water (p10 lines 15-16). Please clarify. What was the drug, supplier, dose and duration of treatment in this experiment?

Response: We apologize for incomplete description in the method section. We revised the method section with more details, and the new method section which now reads:

α MHC-tTA mice were crossed with TRE-DIR/LacZ transgenic mice for generating the cardiomyocyte (CM)-specific DIR-overexpressing mice whose expression of DIR could be naturally induced together with the expression of LacZ reference and this induction is reversed by administration of doxycycline through the drinking water. α MHC-tTA/TRE-DIR mice were treated with 0.2g/l doxycycline (Sigma D-9891) in tap water containing 1% sucrose for two weeks. Doxycycline containing water was shielded from light and was replaced with fresh one

two times per week. (page 11, line14 - page 12, line 2)

- *Figure 1f, how many weeks post-surgery is this?*

Response: It was 4 weeks after TAC surgery. The information can now be found in the figure legend.

2) *Linked to the above, there are statistical concerns regarding the methodology used to compare between groups in single cell/slice comparisons, particularly because it is unclear how many actual mice were used per condition, and how many cells/slices came from each mouse.*

- *Figure 1d, how many mice per timepoint? How many cells analysed per mouse? Why are different numbers of cells analysed for each timepoint, and only 680 for control? When performing the statistical test, should the mean values for each mouse not be calculated first, and then the statistical test performed on these mean values, in order to give a biologically informed comparison?*

Response: We performed smFISH with four different timepoints with each timepoint being representative of one mouse. With regard to the relatively lower number of cells analyzed in the control sample, it is because of WGA staining bias during control conditions. We would like to note that this bias would not affect the assessment of target gene expression but could be the matter for determining the CMs. To address the point raised by the reviewer, we adjusted sample volume bias by downsampling the data, and still found the same result as shown in Figure 1d. Thus, we conclude that our result shown in Figure 1d is correct.

We agree with the reviewer that our data does not contain a biological replicate, but our focus in this experiment is to determine the expression pattern of *Drd1* in the failing heart, and we successfully proved it by using smFISH with our previously developed algorithm. We need to take into account that each experimental technique has pros and cons. The advantage of smFISH is the ability to detect spatial transcriptional information from the tissue section. In addition, the automated system developed by our group allows us to determine over hundreds and thousands of CMs from heart tissue section without selection bias. However, the limitations of this method are that it is not an appropriate system to process multiple samples at once and that there is relatively higher background signal leading to pseudo positive outcomes. In order to compensate for the shortcomings of smFISH data, we performed scRNA-seq in mice and human with biological replicates. By integrating the results of both smFISH and scRNA-seq, we confirmed our finding. We believe that determining the phenomenon by multiple modalities would allow us to overcome the limitation of each modality and to reach the true conclusion. We hope that our argument was able to alleviate the reviewer's concern.

- *Figure 1e, how many mice per condition? How many cells from each mouse?*

Response: We collected rod-shaped viable CMs right after isolation with the Langendorff method from ~48 cells per mouse. For all time points, two mice were subjected to SMART-seq 2 based scRNA-seq. The number of cells shown in the figure varied as a result of quality control during the process of library preparation, where the low-quality cells were filtered out based on the following two steps: the first step was based on an assessment of the efficacy of reverse transcription and amplification using real-time quantitative PCR. The second step was based on an assessment of the number of detected genes after sequencing the single-cell libraries. We followed the details of these quality controls as has been described in our previous paper². We added the information to Figure 1e's figure legend and method section in the revised manuscript.

Figure 1f, as above for 1d. How many mice per condition? How many slices analysed per mouse? Were mean values per mouse used for the stats test. Also, why is a bar chart used here instead of a dot-plot?

Response: With regard to this figure, we initially sought to provide another evidence to ensure the increase in numbers of *Drd1* positive cells in failing hearts by employing the novel *Drd1* knock-in LacZ reporter mouse line, since we had difficulty in detecting D1R protein in the heart tissues (see our response to the concern after next). The advantage of this system is that we can detect endogenous induction of D1R by LacZ reporter staining. The data shown in Figure 1f was acquired from 113 slices from one TAC-operated animal, and 146 slices from two sham controls. As we have closed this mouse line and had a difficulty to re-expand the colony for additional set of TAC experiments in order to obtain enough numbers of animals for analysis in a timely-manner, we now would like to remove the bar graph to avoid any confusion and concerns as you raised. Nevertheless, we would like to keep the representative images with positive staining for LacZ, indicating the endogenous expression of cardiac D1R (Figure 1f and Supplementary figure 2c). We believe these images could be helpful for readers to gain understanding of patchy expression of D1R.

- *Figure 2e, was this experiment reproduced with a second batch of neonatal CMs? Or are all data obtained from one biological experiment? n=9 and n=10 seem small if this refers to numbers of cells analysed. Were the cells randomly selected, was the experimenter blinded to the experimental condition?*

Response: We agree with the reviewer that we should have shown more solid data. The data in Figure 2e was the representative data from one experiment. Note that the observation has been reproducible, but the problem of this experiment is that the efficiency of adenovirus

transduction and functional stability of primary neonatal CMs varies a lot across the experiments. We performed more than three repeats of this experiment and confirmed the reproducibility of the observation. Now, Figure 2e has been revised with larger cell numbers (60 cells per condition).

3) *The study relies upon mRNA expression to infer expression of the D1R protein receptor in CMs. This requires validation of D1R protein expression by immunohistochemistry. There appear to be several antibodies available for this. Authors might co-stain with a CM-specific marker to demonstrate CM-specific protein expression of D1R in mouse and human control and heart failure tissue.*

Response: This is a fair critique. Although we totally agree that protein levels in CMs need to be validated by immunohistochemistry, we have had unsuccessful attempts with several antibodies that are commercially available and widely used for detection of D1R in brains (D2944, Sigma; Ab20066, Abcam; SG2-D1a, Novus). We speculate that this could be attributable to the extremely low levels of D1R protein expression in CMs compared to brains, even only a small number of CMs are positive for D1R in failing hearts. We checked *Drd1* transcript levels between hearts and brains using publicly available transcriptome data (mouse data (BioProject: PRJNA66167, PMID 25409824); human data (BioProject: PRJNA280600, PMID 25970244)), and found that there is at least 50-100 times difference between hearts and brains.

These results, which might support our speculation, have been provided in Figure R3-1. Next, we sought to detect D1R protein at heart tissues by western blot analysis with whole heart tissues from control and TAC-operated mice, however, we failed to detect D1R protein even with failing hearts. To enrich the D1R protein that express on the cellular membrane, we fractionated the membrane protein from heart tissues and managed to detect a weak signal for D1R from TAC-operated hearts, but not from control hearts, while strong bands were observed in brains (using D2944 antibody, Ref 12). We have provided this result only to this reviewer given

Figure R3-1: *Drd1/DRD1* mRNA expression in heart and brain tissues. *Drd1* mRNA expression in heart and brain tissues in mice (left). *DRD1* mRNA expression in human hearts and brains (right).

Figure R3-2: D1R protein expression in mouse heart and brain tissues by western blotting. D1R expression was determined with the membrane fraction from sham heart tissues, TAC-operated heart tissues, and brain tissues in mice. D1R was barely detected on sham heart but slightly on TAC heart. In contrast, D1R was highly expressed in brain tissue.

the preliminary nature of the results (Figure R3-2). Instead, we cited the previous reports showing the existence of DIR in CMs (Ref 24-26) in the main text.

4) It is claimed that figures 1d and 1f demonstrate DIR expression in CMs. However, how do the authors discount the possibility that some +ve stained cells are non-CMs? For example, DIR expression in vasculature is known. Authors should either quantify only DIR+ staining in cells that are co-stained with a CM-specific marker, or, clearly acknowledge the caveat that some staining here may arise from non-CM cells.

Response: We thank the reviewer for this insightful comment. We would like to argue that our algorithm with smFISH based assessment has been validated for its specificity to detect CMs by examining CM specific gene expression (*Atp2a2*, *Tnnt2*) and non-CM reference gene expression (endothelial reference marker genes *Pecam1* and *Cav1*; fibroblast reference marker genes *Lum* and *Dcn*)³. All the cells defined as CMs by our algorithm was both *Atp2a2*- and *Tnnt2*-positive but negative for *Pecam1*, *Cav1*, *Lum*, or *Dcn*. We have clarified this point in the main text of revised manuscript (page5, line6).

5) Doxycycline is a known cardioprotective agent [PMIDs 24104875, 31274354]. There is therefore concern that reduced ventricular arrhythmia following DOX administration (Fig 2b) could be confounded by the effect of DOX itself, rather than reduced DIR expression. An ideal solution could be to repeat this experiment using a Tet-ON system (*aMHC-rtTA*). However, alternatively, authors might demonstrate that DOX administration alone does not rescue ventricular arrhythmia in wild-type mice after TAC (i.e. figure 1g experimental setup).

Response: We thank the reviewer for raising this point. Due to time constraints for this revision cycle, we decided to adopt the alternative experiment as suggested by the reviewer. We determined the pleiotropic effect of DOX on PVCs in our TAC model with wild-type mice, and found that DOX administration didn't affect the frequency of PVCs after TAC (Figure R3-3, Supplementary Figure 4c). This data indicates that the cardioprotective effect of DOX is negligible on TAC induced ventricular arrhythmia and likely not be a confounding factor in our assessment.

Figure R3-3: Impact of DOX treatment on TAC-induced heart failure and arrhythmia. (Left) Wild-type mice after 6 weeks of TAC operation were treated with or without 2 weeks DOX treatment and assessed for the frequency of caffeine/dopamine-induced PVCs. (Middle and Right) Cardiac function of pre-DOX treatment. There was no significant difference in the frequency of PVCs by DOX treatment.

Minor:

1) *Were there any differences in DIR KO body weights, or heart weights?*

Response: As per reviewer's request, we compared the body weights and heart weights between $\alpha MHC-Cre$ mice and $\alpha MHC-Cre/Drd1^{fl/fl}$ mice. There was no statistical difference between them, but we observed a trend that $\alpha MHC-Cre/Drd1^{fl/fl}$ mice is a bit lighter than $\alpha MHC-Cre$ mice. The data has been included in Supplementary Table 1-2.

Also, were there differences in heart contractile function, wall thickness, heart weight or body weight in the DIR overexpression model?

Response: As requested, we checked those parameters and compared between $\alpha MHC-tTA$ and $\alpha MHC-tTA/TRE-DIR$ mice. We didn't detect the difference between two groups and the data is now available in Supplementary Table 1-3.

Note that dosage of the caffeine-dopamine intravenous drug infusion was not adjusted for body weight.

Response: As the reviewer pointed out, we haven't adjusted the speed of caffeine-dopamine intravenous infusion based on the body weight. There is a technical reason why we could not adjust the speed. Due to concern of volume overload with infusion using a normal infusion pump, we adopted a very low-speed infusion pump with the range of speed with 0.1-2ul/min. The downside of this method is that we have to use concentrated caffeine-dopamine reagent. Since the gap among mice was +/- 5-10%, it was too small for this pump to adjust the flow rate based on the body weight. As we shown above, there was no difference in body weight between case and control. Furthermore, the body weight of $\alpha MHC-Cre/Drd1^{fl/fl}$ mice was smaller than control group, indicating that $\alpha MHC-Cre/Drd1^{fl/fl}$ mice received higher dose of caffeine-dopamine mix per weight, even though the frequency of TAC-induced PVCs were significantly suppressed in $\alpha MHC-Cre/Drd1^{fl/fl}$ mice. Thus, although the reviewer has a point and we cannot ignore the potential bias based on the body weight difference, the overall conclusion led by our data would not be affected by this issue.

2) *Figure 1f: On average, how many total CMs are there per slice? 0.3 LacZ+ cells per slice (TAC condition) seems low. Does this correspond to the proportion of DIR+ CMs identified by scRNAseq (from Figure 1e, for 4 week post-TAC, this proportion is 5/73 = 6.8%)?*

The representative image in figure 1f and supplementary figure 2c suggest that fewer than 6.8% of CMs are LacZ+. Can the authors comment on this discrepancy?

Response: We think this is a good point and we agree that there is a discrepancy between X-gal staining based LacZ detection and transcriptomic based *Drd1* mRNA detection (smFISH and

scRNA-seq). Our X-gal staining protocol might cause this, because we titrated the staining protocol, resulted in shorter incubation time to acquire specific signal. Also, this could be explained by the difference of each detection system. As for X-gal staining on *DIR-tTA/TRE-DIR/LacZ* mice, there are a couple of steps until LacZ protein is generated from transactivation of *DIR* gene to its enzymatic activity is assessed. In contrast, scRNA-seq/smFISH can directly assess *DIR* mRNA itself, so it's more straight-forward. Given these facts, it is not surprising to see the discrepancy between them, but the point here is that we could detect the *DIR* expressing CMs by using multiple different methods.

3) *Supplementary Figure 3c: No error bars, was this n=1? Also, was this measured using an un-operated mouse? If so, it's surprising that such a clear difference exists, given that in un-operated mice, almost no CMs are expected to express DIR. Can the authors comment on this?*

Response: This experiment was done with Langendorff-isolated CMs from TAC mouse (n=1 per genotype). We performed additional experiments with more mice from both genotypes, and found that the expression of *Drd1* in the failing heart of cardiac specific *DIR* CKO mice was significantly lower than that of control genotype (Supplementary Figure 3c). As the reviewer noticed, we could not detect the difference of *Drd1* between CKO and control at the sham condition.

4) *P4 Line 14 "Increased expression of Drd1, but no other dopamine receptor genes, was confirmed by quantitative PCR (qPCR)." Only qPCR for DIR and D2R are shown – what about others?*

Response: As indicated above under reviewer #1 (comment #1), we performed additional qPCR and the expression data for *Drd3-5* expression has been included in Supplementary Figure 1.

5) *Please show primer sequences used for qPCR.*

Response: The primer sequences and UPL probes for qPCR are now available in Supplementary Table 3.

Reference

- 1 Hampton, J. R. *et al.* Randomised study of effect of ibopamine on survival in patients with advanced severe heart failure. Second Prospective Randomised Study of Ibopamine on Mortality and Efficacy (PRIME II) Investigators. *Lancet* **349**, 971-977 (1997).
- 2 Nomura, S. *et al.* Cardiomyocyte gene programs encoding morphological and functional signatures in cardiac hypertrophy and failure. *Nat Commun* **9**, 4435, doi:10.1038/s41467-018-06639-7 (2018).
- 3 Satoh, M. *et al.* High-throughput single-molecule RNA imaging analysis reveals heterogeneous responses of cardiomyocytes to hemodynamic overload. *J Mol Cell Cardiol* **128**, 77-89, doi:10.1016/j.yjmcc.2018.12.018 (2019).

REVIEWERS' COMMENTS:

Reviewer #1 (Remarks to the Author):

The additional studies markedly improved the revised report. Some minor concerns remain.

1. Please define “n/a” and “e” in Supplementary Figure 1a.
2. Lines 53-56. Reference #6 deals with shock, not heart failure.
3. Line 64. The word “algorism” is not the correct word. The authors probably meant “algorithm”. According to Wikipedia, informally, an algorithm can be called a "list of steps". By contrast, “algorism is the technique of performing basic arithmetic by writing numbers in place value form and applying a set of memorized rules and facts to the digits” (according to Wikipedia).
4. Lines 67-69. Reference 11 cited in this sentence is missing the name of the journal (Nature Communications), line 539.
5. Line 89. Epinephrine, not dopamine, is used with caffeine in reference #13.
6. Lines 95-72. The phrase “lesser mortality” may be better than “better mortality” in the context of the sentence.
7. Supplementary Tables 1-1 to 1-3. LVFS: fractional is spelled as “fractinal”.
8. Lines 144-146. Although a couple of papers suggested the existence of D1R in human and rat hearts, there have been no reports addressing the physiological and pathophysiological function of cardiac D1R. The authors cited three articles. Therefore, the phrase “couple of papers” is not appropriate.
9. The word “labeling” (line 148) is also spelled “labelling” (line 69).
10. The immunoblot analysis section does not include antibody validation.
11. The D1R gene in humans is DRD1 (line 334); it is Drd1 in non-humans.

Reviewer #2 (Remarks to the Author):

I am pleased with the responsiveness of the authors. Significant new data and discussion has been added.

Reviewer #3 (Remarks to the Author):

The authors have completed a thorough rebuttal. My concerns have been addressed.

Just a few minor points below, to clarify some areas of the manuscript.

1. Fig 1d and 2e. Please clarify in the figure legends that these were from n=1 mice per group.
2. Fig 1d. Is “week 0” actually a sham-operated heart, as currently stated in the legend? Or is this a pre-operation heart. Please clarify.
3. Fig 1d. Please show on the figure, is there statistical significance, between sham group and any of the other time points (1 week, 2 weeks)?
4. Please add a line in the Methods section to clarify that age differences (i.e. between 8-12 week old mice) were limited by using littermate controls.
5. Methods section, Line 194, “and was replaced with fresh one two times per week” this is slightly confusing wording, perhaps authors can replace with “and was refreshed twice per week”.

Response to referees

First, we thank all the reviewers for the interests in our paper and thoughtful comments again. We have addressed all comments in a point-by-point fashion as below. Based on the comments and suggestions, we have checked all figures and analyses and cleared up some confusion. All changes within the manuscript is highlighted with green markings. Overall, we think that the revised manuscript becomes more clear and solid. Our responses to the reviewer's comments are marked with red text. We wish to thank all for their attention to details and for their kind comments.

REVIEWERS' COMMENTS:

Reviewer #1 (Remarks to the Author):

The additional studies markedly improved the revised report. Some minor concerns remain.

1. Please define “n/a” and “e” in Supplementary Figure 1a.

We defined both in the legend.

2. Lines 53-56. Reference #6 deals with shock, not heart failure.

We agree with the reviewer. We excluded this reference in the revised manuscript.

3. Line 64. The word “algorism” is not the correct word. The authors probably meant “algorithm”. According to Wikipedia, informally, an algorithm can be called a "list of steps". By contrast, “algorism is the technique of performing basic arithmetic by writing numbers in place value form and applying a set of memorized rules and facts to the digits” (according to Wikipedia).

We thank the reviewer for the advice. We corrected the spelling to “algorithm”.

4. Lines 67-69. Reference 11 cited in this sentence is missing the name of the journal (Nature Communications), line 539.

We corrected the reference.

5. Line 89. Epinephrine, not dopamine, is used with caffeine in reference #13.

We thank the reviewer for the advice. We excluded this reference in the revised manuscript.

6. Lines 95-72. The phrase “lesser mortality” may be better than “better mortality” in the

context of the sentence.

We corrected the phrase to “lesser mortality”.

7. Supplementary Tables 1-1 to 1-3. LVFS: fractional is spelled as “fractinal”.

We corrected the typo.

8. Lines 144-146. Although a couple of papers suggested the existence of D1R in human and rat hearts, there have been no reports addressing the physiological and pathophysiological function of cardiac D1R. The authors cited three articles. Therefore, the phrase “couple of papers” is not appropriate.

We thank the reviewer for the advice. We changed the phrase “couple of papers” to “a few papers”.

9. The word “labeling” (line 148) is also spelled “labelling” (line 69).

We corrected “labelling” to “labeling”.

10. The immunoblot analysis section does not include antibody validation.

We thank the reviewer for raising the point. We did not perform antibody validation but instead, we have added the position of molecular weight markers in Figure 3f (previous Figure 2f), added uncropped images of the immunoblot as Supplementary Figure 6, and noted the reference paper about the antibody used in this study. We contacted the authors of the reference paper and were advised not to perform heat-denaturation before SDS-PAGE, because it could disturb the recognition of target site by the antibody. We also described this point in Methods section.

11. The D1R gene in humans is DRD1 (line 334); it is Drd1 in non-humans.

We have corrected Drd1 to DRD1.

Reviewer #2 (Remarks to the Author):

I am pleased with the responsiveness of the authors. Significant new data and discussion has been added.

We thank the reviewer for the favorable comment.

Reviewer #3 (Remarks to the Author):

The authors have completed a thorough rebuttal. My concerns have been addressed.

Just a few minor points below, to clarify some areas of the manuscript.

1. Fig 1d and 2e. Please clarify in the figure legends that these were from n=1 mice per group.

As suggested, we added the number of the animals used for the experiments in legend for Fig. 1d and in Fig. 2e (now Fig. 3e).

2. Fig 1d. Is “week 0” actually a sham-operated heart, as currently stated in the legend? Or is this a pre-operation heart. Please clarify.

As suggested by the reviewer, this is “sham-operated” heart. We therefore added that “0 week” heart means “sham-operated” heart.

3. Fig 1d. Please show on the figure, is there statistical significance, between sham group and any of the other time points (1 week, 2 weeks)?

We thank the reviewer for the comment. Yes, we observed statistically significant difference between 1 week, but the presence of 2 outlier cells that highly express Drd1 made the difference not significant ($p > 0.9999$). Cells that highly express Drd1 increased more at 8 weeks, making the difference statistically significant at that time point. We added this information in Fig. 1d.

4. Please add a line in the Methods section to clarify that age differences (i.e. between 8-12 week old mice) were limited by using littermate controls.

We added sentences in Methods section accordingly (page 13, line 7-8).

5. Methods section, Line 194, “and was replaced with fresh one two times per week” this is slightly confusing wording, perhaps authors can replace with “and was refreshed twice per week”.

We thank the reviewer for pointing out our confusing wording. We corrected the text accordingly.